# Structural analysis of rice Os4BGlu18 monolignol β-glucosidase

**Supaporn Baiya[1], Salila Pengthaisong[2], Sunan Kitjaruwankul[1], James R. Ketudat Cairns[2]***

**1** Faculty of Science at Sriracha, Kasetsart University, Sriracha Campus, Sriracha, Chonburi, Thailand,
**2** School of Chemistry, Institute of Science and Center for Biomolecular Structure, Function and Application, Suranaree University of Technology, Muang, Nakhon Ratchasima, Thailand

* cairns@sut.ac.th

## Abstract

Monolignol glucosides are storage forms of monolignols, which are polymerized to lignin to strengthen plant cell walls. The conversion of monolignol glucosides to monolignols is catalyzed by monolignol β-glucosidases. Rice Os4BGlu18 β-glucosidase catalyzes hydrolysis of the monolignol glucosides, coniferin, syringin, and *p*-coumaryl alcohol glucoside more efficiently than other natural substrates. To understand more clearly the basis for substrate specificity of a monolignol β-glucosidase, the structure of Os4BGlu18 was determined by X-ray crystallography. Crystals of Os4BGlu18 and its complex with δ-gluconolactone diffracted to 1.7 and 2.1 Å resolution, respectively. Two protein molecules were found in the asymmetric unit of the $P2_12_12_1$ space group of their isomorphous crystals. The Os4BGlu18 structure exhibited the typical $(β/α)_8$ TIM barrel of glycoside hydrolase family 1 (GH1), but the four variable loops and two disulfide bonds appeared significantly different from other known structures of GH1 β-glucosidases. Molecular docking studies of the Os4BGlu18 structure with monolignol substrate ligands placed the glycone in a similar position to the δ-gluconolactone in the complex structure and revealed the interactions between protein and ligands. Molecular docking, multiple sequence alignment, and homology modeling identified amino acid residues at the aglycone-binding site involved in substrate specificity for monolignol β-glucosides. Thus, the structural basis of substrate recognition and hydrolysis by monolignol β-glucosidases was elucidated.

## Introduction

Lignin has biological functions of strengthening and waterproofing plant cell walls, which differ in mechanical properties in specific cell types [1]. It is a natural polyphenolic polymer composed of the *p*-hydroxyphenyl (H), guaiacyl (G), and syringyl (S) units, which are derived from oxidative polymerization of the corresponding monolignols, *p*-coumaryl alcohol, coniferyl alcohol, and sinapyl alcohol, respectively [2]. It is well-known that monolignols are synthesized via the phenylalanine pathway and either directly secreted to the cell wall or glucosylated by uridine diphosphate glucosyltransferase to generate monolignol glucosides, including *p*-coumaryl alcohol glucoside, coniferin, and syringin [3–6]. These monolignol glucosides are mostly

**Data Availability Statement:** The X-ray crystal structure data in this manuscript, including structure files and structure factors, are available from the Protein Data Bank, PDB entries 7D6A and 7D6B. The authors submitted their data to PDBj

and can be found at the following: https://pdbj.org/mine/summary/7D6A and https://pdbj.org/mine/summary/7D6B.

**Funding:** Data collection was supported by grants from SPring-8 and the Institute for Protein Research, Osaka University (http://www.protein.osaka-u.ac.jp/en/). Data collection at the NSRRC was supported by Synchrotron Radiation Protein Crystallography Facility of the National Core Facility Program for Biotechnology, Ministry of Science and Technology and the NSRRC, a national user facility supported by the Ministry of Science and Technology, Taiwan, ROC. This work was supported by the Thailand Research Fund (trf.or.th/eng/) and Suranaree University of Technology (SUT, www.sut.ac.th) (grants BRG5980015 and RSA6280073). SB was partially supported by the Faculty of Science at Sriracha Kasetsart University. SP was supported by a full-time postdoctoral researcher grant from SUT and Thailand Science Research and Innovation (TSRI).

**Competing interests:** The authors have declared that no competing interests exist.

found in the lignifying secondary cell wall of gymnosperms and are expected to be storage forms of monolignols, although they do not seem to be required for lignin synthesis [4, 7–12].

To release monolignol glucosides, monolignol β-glucosidases catalyzed the hydrolysis of the glycosidic bond between nonreducing β-D-glucosyl residues and the corresponding free aglycone [13, 14]. They belong to glycoside hydrolase family 1 (GH1), members of which act in many essential roles, including hydrolysis of structural or storage polysaccharides, defense against pathogens, phytohormone release from storage glycosides, and turnover of cell surface carbohydrates, etc [15]. GH1 members catalyze their reactions with a molecular mechanism leading to overall retention of the anomeric configuration, which involves the formation and breakdown of a covalent glycosyl enzyme intermediate [16]. All of the GH1 enzymes display a common $(\beta/\alpha)_8$ TIM barrel structure and contain two conserved catalytic glutamate residues located at the C-terminal end of β-strands 4 and 7 [17, 18]. There have been a number of GH1 crystal structures from plants, such as *Zea mays* [19], *Sorghum bicolor* [20], and *Oryza sativa* [21], produced to explain substrate specificity, based on the bonding between the amino acid residues around the enzyme active site and its substrate [15].

Rice Os4BGlu18 is a monolignol β-glucosidase which exhibited high substrate specificity for the monolignol glucosides, coniferin ($k_{cat}/K_m$, 31.9 mM$^{-1}$s$^{-1}$), syringin ($k_{cat}/K_m$, 24.0 mM$^{-1}$s$^{-1}$), and *p*-coumaryl alcohol glucoside ($k_{cat}/K_m$, 1.4 mM$^{-1}$s$^{-1}$) [14]. Moreover, *Arabidopsis thaliana bglu45* mutant rescued by expression of Os4BGlu18 displayed lower levels for all three substrates when compared with non-rescued line [12]. Since there is no published structure for a monolignol β-glucosidase, the aim of this work was to determine the structure of Os4BGlu18 monolignol β-glucosidase and investigate its interactions with monolignol glucosides via computational docking.

## Materials and methods

### Protein expression, purification, and crystallization

The recombinant N-terminally thioredoxin and His$_6$-tagged Os4BGlu18 protein was expressed and purified by immobilized metal affinity chromatography (IMAC), as previously described [14, 22]. The N-terminal thioredoxin-His$_6$-tagged was cleaved with 2 ng enterokinase per milligram of protein at 23°C for 16 h, then a second round of IMAC was done to remove the tag. The purified Os4BGlu18 protein was dialyzed and concentrated with a 10 kDa Amicon® Ultra-15 centrifugal filter. To screen for Os4BGlu18 protein crystallization conditions, the protein was diluted to 3 mg/ml with a buffer containing 20 mM Tris-HCl, pH 8.0, 150 mM NaCl. The initial crystallization screens of Os4BGlu18 were set up by the microbatch under oil method with precipitants from the Crystal Screen HT kit (Hampton Research), by mixing the well-solution and protein sample at a 1:2 volume ratio. The crystallization was optimized by hanging drop vapor diffusion method, varying the concentrations of polyethylene glycol (PEG) 3350 over the range of 20–24%, and protein between 1 and 8 mg/ml in 100 mM MES, pH 6.0, at 288 K. Upon optimization, the high quality crystals were produced in precipitants consisting of 21% PEG 3350, 100 mM MES, pH 6.0. Prior to data collection, the crystals were transferred to precipitant supplemented with 18% (v/v) glycerol (cryoprotectant) and flash-vitrified in liquid nitrogen. Crystals of the Os4BGlu18 were soaked in 1 mM δ–gluconolactone (DG) in cryoprotectant for 20 min before vitrification to obtain the Os4BGlu18-DG complex.

### Data collection, processing and structure refinement

The crystals without and with DG were initially diffracted at the SPring-8 synchrotron beamline BL44XL with 0.9000 Å X-ray radiation on a Rayonix MX-300HE detector. The crystals

gave low resolution diffraction, therefore, higher quality of crystals were produced. The X-ray diffraction data were collected at beamline BL15A1, National Synchrotron Radiation Research Center (NSRRC, Hsinchu, Taiwan), with a wavelength of 1.00 Å and a complete dataset of 180 1˚-images was collected on a Rayonix MX300HE CCD detector. The diffraction images were processed and scaled by the program HKL2000 [23]. The crystals of Os4BGlu18 and Os4B-Glu18-DG complex diffracted to 1.7 and 2.1 Å, respectively, and had the orthorhombic space group $P2_12_12_1$. Refinement statistics are summarized in Table 1. Molecular replacement of Os4BGlu18 was carried out by *MOLREP* in the *CCP4* suite [24] using *Oryza sativa* Os3BGlu7

**Table 1. Data collection and refinement statistics.**

| Dataset | Os4BGlu18 | Os4BGlu18-DG complex |
|---|---|---|
| PDB code | 7D6A | 7D6B |
| Beamline | BL15A1 | BL15A1 |
| Wavelength (Å) | 1.00 | 1.00 |
| **Data collection** | | |
| Space group | $P2_12_12_1$ | $P2_12_12_1$ |
| Unit-cell parameter (Å) | $a = 52.1$ | $a = 52.1$ |
| | $b = 83.8$ | $b = 83.8$ |
| | $c = 207.6$ | $c = 207.4$ |
| Resolution range (Å) | 40.0–1.70 | 40.0–2.10 |
| Resolution outer shell (Å) | 1.76–1.70 | 2.18–2.10 |
| No. Unique reflections | 96808 | 53787 |
| No. Observed reflections | 520913 | 352015 |
| Completeness (%) | 96.0 (97.0) | 98.9 (99.5) |
| Average redundancy per shell | 5.4 (4.6) | 6.6 (5.3) |
| I/σ (I) | 14.3 (2.9) | 23.2 (12.3) |
| $R_{(merge)}$ (%) | 6.2 (48.3) | 5.9 (12.5) |
| CC1/2 | (0.841) | (0.987) |
| **Refinement** | | |
| $R_{factor}$ (%) | 15.1 | 14.7 |
| $R_{free}$ (%) | 17.5 | 17.7 |
| No. of residues in proteins | 950 | 950 |
| No. Protein atoms | 3887/3889 | 3867/3877 |
| No. Ligand atoms | None | 12/12 |
| No. Other hetero atoms | 103 | 97 |
| No. Waters | 831 | 613 |
| **Mean B-factor** | | |
| Protein | 18.3/18.0 | 19.5/19.5 |
| Ligand | None | 14.4/15.4 |
| Other hetero atoms | 28.7 | 37.1 |
| Waters | 29.2 | 26.8 |
| r.m.s. bond deviations (length) | 0.007 | 0.008 |
| r.m.s angle deviations (degrees) | 1.38 | 1.35 |
| **Ramachandran plot** | | |
| Residues in most favorable regions (%) | 89.1 | 89.7 |
| Residues in additional allowed regions (%) | 9.7 | 8.7 |
| Residues in generously allowed regions (%) | 1.0 | 1.1 |
| Residues in disallowed regions (%) | 0.2 | 0.5 |

(PDB code 2RGL; [25]) as the search model. The final model was refined and rebuilt with *REFMAC5* and *WinCoot* software, respectively [26, 27]. The structure of the Os4BGlu18-DG complex was solved by rigid-body refinement of the native Os4BGlu18 structure. The ligand was derived from RCSB Protein Data Bank (https://www.rcsb.org/) and refined to the density map. Stereochemical analysis was performed using *PROCHECK* [28] and *MolProbity* [29]. Finally, *PyMOL* software was used to visualize the structure. The one metal ion observed in the structure was modeled as zinc, based on Zn giving the best match to the density and lowest B-factor, and was validated with the CheckMyMetal server (https://cmm.minorlab.org/) [30].

## Docking and modeling

Molecular docking was performed using Autodock 4.2 (ADT version 1.5.6) to investigate the interactions of Os4BGlu18 with its natural substrates. All hetero atoms and molecule B residues were deleted from the native Os4BGlu18 structure. Polar hydrogen atoms and Kollman charges were added to the protein. A hydrogen atom was manually added to the catalytic acid/base, E194, to be consistent with its role in protonating the glycosidic oxygen in the first step of catalysis. The *p*-coumaryl alcohol glucoside, coniferin, and syringin ligands were prepared with the glucosyl moiety in the $^1S_3$ skew boat conformation with Discovery Studio 4.0 program (Dassault Systèmes BIOVIA, San Diego, CA) and set the number of active torsions as 6, 7, and 8, respectively. The aromaticity criterion was set to be 7.5. The ligands were docked into the active site of Os4BGlu18 with the grid box dimension 60x60x60 points in x, y, and z and a grid spacing of 0.375 Å *via* a Lamarkian genetic algorithm methodology. The docking was run 200 times for each substrate using Cygwin and the conformation which showed the best binding energy (ΔG) was selected [31].

Homology modeling of the three-dimensional structure of Os4BGlu14, Os4BGlu16, AtBGlu45, AtBGlu46, and *Pinus contorta* coniferin β-glucosidase (PcBGlu) were obtained using SWISS-MODEL server (https://swissmodel.expasy.org/) with the structure of Os4BGlu18 as a template.

MD simulations of a docked protein-ligand complex (protein-coniferin) was carried out using NAMD software [32]. The CHARMM36m force field [33] was used for protein, whereas the force field of coniferin was calculated via the charmm-gui server [34]. The PSF file was generated using Visual molecular dynamics (VMD) [35]. The docked system was solvated in a cubic water box containing a transferable intermolecular potential with 3 points (TIP3P) water molecules [36]. The box size was chosen so that there was a distance of 15 Å between the protein surface and the edges of the periodic box with a size of 90x90x90 Å$^3$. The system contained a total of 70,069 atoms. A cut-off radius for non-bonded interactions was calculated at 12 Å. The particle mesh Ewald (PME) [37] method was used to calculate long-range electrostatic interactions. The SHAKE algorithm [38] was used to constrain all bonds involving hydrogen atoms. The system was first minimized for 50,000 steps of steepest descent, then heated from 50 K to 300 K while restraining protein backbone and ligand molecule, after that the protein backbone and ligand molecule was gradually released and equilibrated at 300 K for 5 ns. The production MD run was performed using an NPT ensemble. The Nosé–Hoover method [39] was used to maintain a constant temperature. The simulation time step was set to 2 fs. The studied simulation time was 20 ns.

## Results and discussion

### Overall and active site structure of Os4BGlu18

Single crystals of Os4BGlu18 and its complex with DG yielded X-ray diffraction datasets with 1.7 and 2.1 Å resolution, respectively (Table 1). The crystals were isomorphous with two

monomeric protein molecules in the asymmetric unit of the $P2_12_12_1$ space group with a Matthews coefficient ($V_m$) of 2.02 Å$^3$ Da$^{-1}$ and 39% (v/v) solvent content within the cell. The Os4BGlu18 structures exhibited the typical $(\beta/\alpha)_8$ TIM barrel found in other known structures of GH1 β-glucosidases [25, 40–50] (Fig 1A). The general acid/base for catalysis is Glu194, which

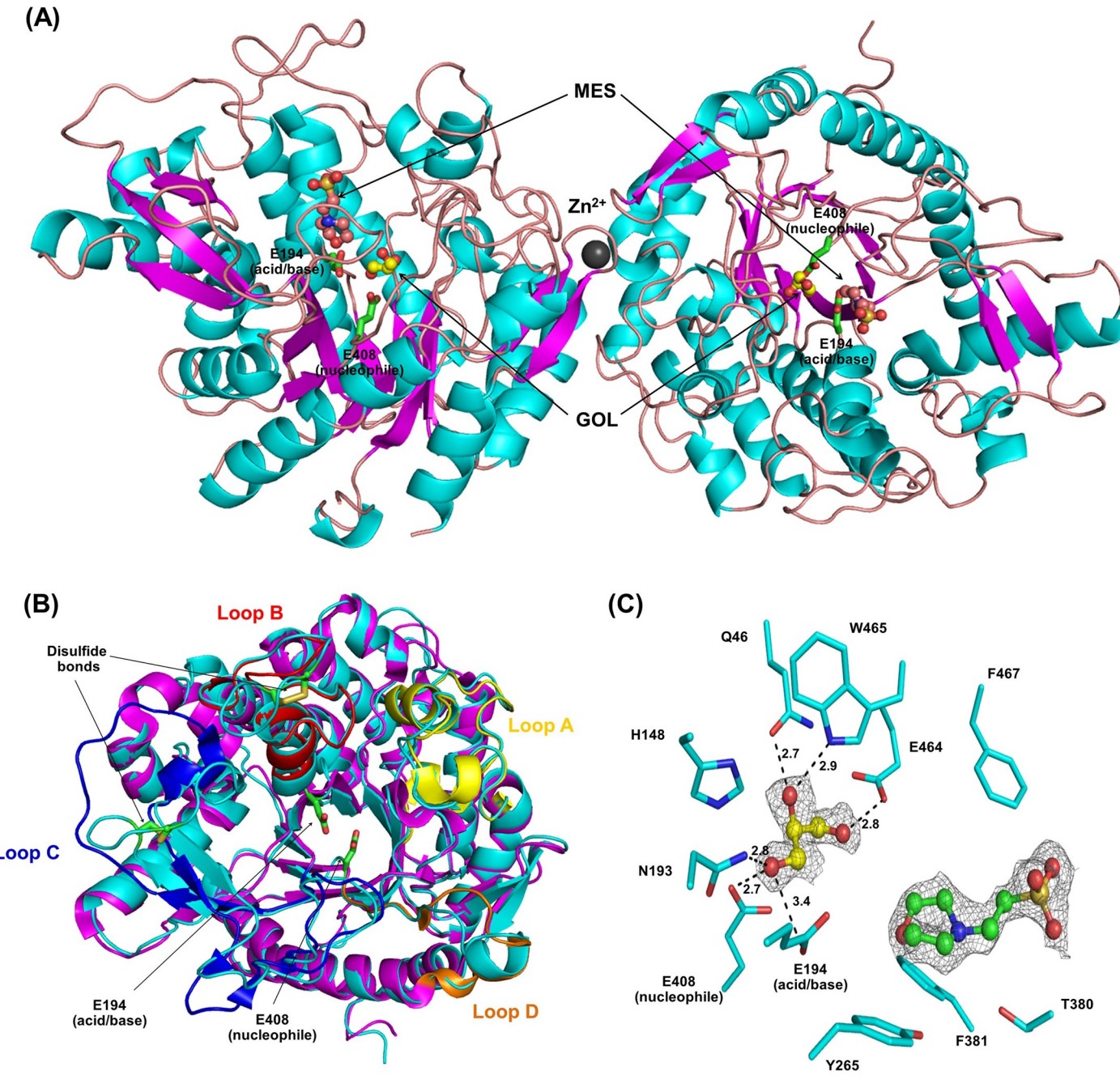

**Fig 1. Asymmetric unit of the native rice Os4BGlu18 structure.** (A) A ribbon diagram exhibiting the overall structure of rice Os4BGlu18 asymmetric unit. The β-strands are colored magenta, α-helices cyan and loops dark salmon. Hetero atoms appeared around the active site of the structure: GOL (glycerols, yellow carbons) and MES (green carbons) are drawn as ball and stick and a dark gray sphere represents Zn$^{2+}$. (B) Superimposition of native Os4BGlu18 structure (cyan) and rice Os3BGlu7 (magenta). Loops A through D of the structure of Os3BGlu7 are colored yellow, red, blue, and orange, respectively. (C) Close-up of the GOL and MES in the active site pocket. Putative hydrogen bonds between these ligands and amino acid residues, based on distances between donor and acceptor of 2.7 to 3.5 Å, are indicated by black dashed lines labeled with distances in Å.

is located in the conserved TFNEP motif found in strand β4 of the barrel. The nucleophile residue, Glu408, is present in the ITENG motif at the C-terminal end of strand β7, consistent with Os4BGlu18 following the double displacement mechanism described for GH1 [16]. The electron density for residues 26–500 of Os4BGlu18 was observed for molecule A for both native and DG complex structures, while density for residues 24–500 was evident for molecule B. These residues correspond to residues 27–500 of Os4BGlu18 and a few residues from the fusion tag, since Os4BGlu18 residues 1–26 are the predicted signal peptide that is cleaved from the protein during synthesis in the plant and were thus not included in the expressed protein. No clear density was observed for the last five amino acid residues at the C-terminus (LHENQ). The two protein molecules are linked by a single metal ion, which was best modeled by zinc, as seen in rice Os3BGlu7 (BGlu1) and Os4BGlu12 GH1 structures [25, 41]. This $Zn^{2+}$ ion was evidentally a minor contaminant in the sodium chloride used in the protein solution, as was also likely in the Os3BGlu7 and Os4BGlu12 structures.

The superimposition between Os4BGlu18 and rice Os3BGlu7 have the same overall structure, however, at the four variable loops around the active site entrance are significantly different (Fig 1B). Loop A residues 72 to 74 (KDG) have a loop structure in Os4BGlu18, while in Os3BGlu7 they have α-helical structure. Loops B and D of Os4BGlu18 are longer than in Os3BGlu7, with insertions at residues 215 to 219 (PPFGH) and 415 to 422 (GDSYTDAE), respectively. Loop C has been noted to be less conserved [45], is more divergent than the other loops, with residues 344 to 357 running the opposite direction of the corresponding residues in Os3BGlu7. Poor density was observed in the side chains of residues 415 to 419 in loop D, whereas for other structures of rice β-glucosidases exhibited poor density in loop C. In Os4BGlu12, loop C is also in the entrance to the active site and was found to be proteolyzed in the protein purified from rice, possibly modifying the substrate specificity [41]. One disulfide bond in loop B is conserved for plant GH1 enzymes, which was found between C213 and C220 in Os4BGlu18, and another disulfide bond was found in loop C (C345 to C350). Two disulfide bonds were also observed in the Os4BGlu12 structure in loop B and at the beginning of loop B connected with α-helix 4 of the TIM barrel. The presence of two disulfide bridges has been suggested to stabilize the enzyme activity at high temperatures [41, 46]. All of these structural differences could contribute to the specificity of GH1 enzymes to their natural substrates.

A glycerol molecule from the cryoprotectant formed hydrogen bonds to the catalytic residues E194 (acid/base) and E408 (nucleophile), and other residues around the active site cleft, including Q46, N193, E464, and W465. The distance between the delta carbon atoms of the E194 and E408 was measured at 5.3 Å, similar to the distance between acid/base and nucleophile in other GH1 structures [19, 20, 25, 40–50]. In addition, a MES molecule from the crystallization buffer was observed at the entrance of the active site pocket of the native Os4BGlu18 structure, although it had no direct polar interactions with the surrounding amino acid residues (Fig 1C).

## The complex of Os4BGlu18 with δ-gluconolactone and structural comparison with other GH1 structures

To study the glycone binding site, several common inhibitors and monolignol glucosides were soaked into the crystal, but clear density was only found for DG. DG is a glucose-like inhibitor that mimiks the expected oxocarbenium ion-like transition state in that the C1 carbon is $sp^2$ hybridized [51], and could be observed in the active site pocket. The Os4BGlu18-DG complex structure contains DG in the ring-form in a $^4H_3/^4E$ conformation (Cremer-Pople parameters: $\varphi$ (˚), $\theta$ (˚), $Q$: 224.521, 47.426, 0.534 and 227.030, 44.337, 0.524 in molecules A and B,

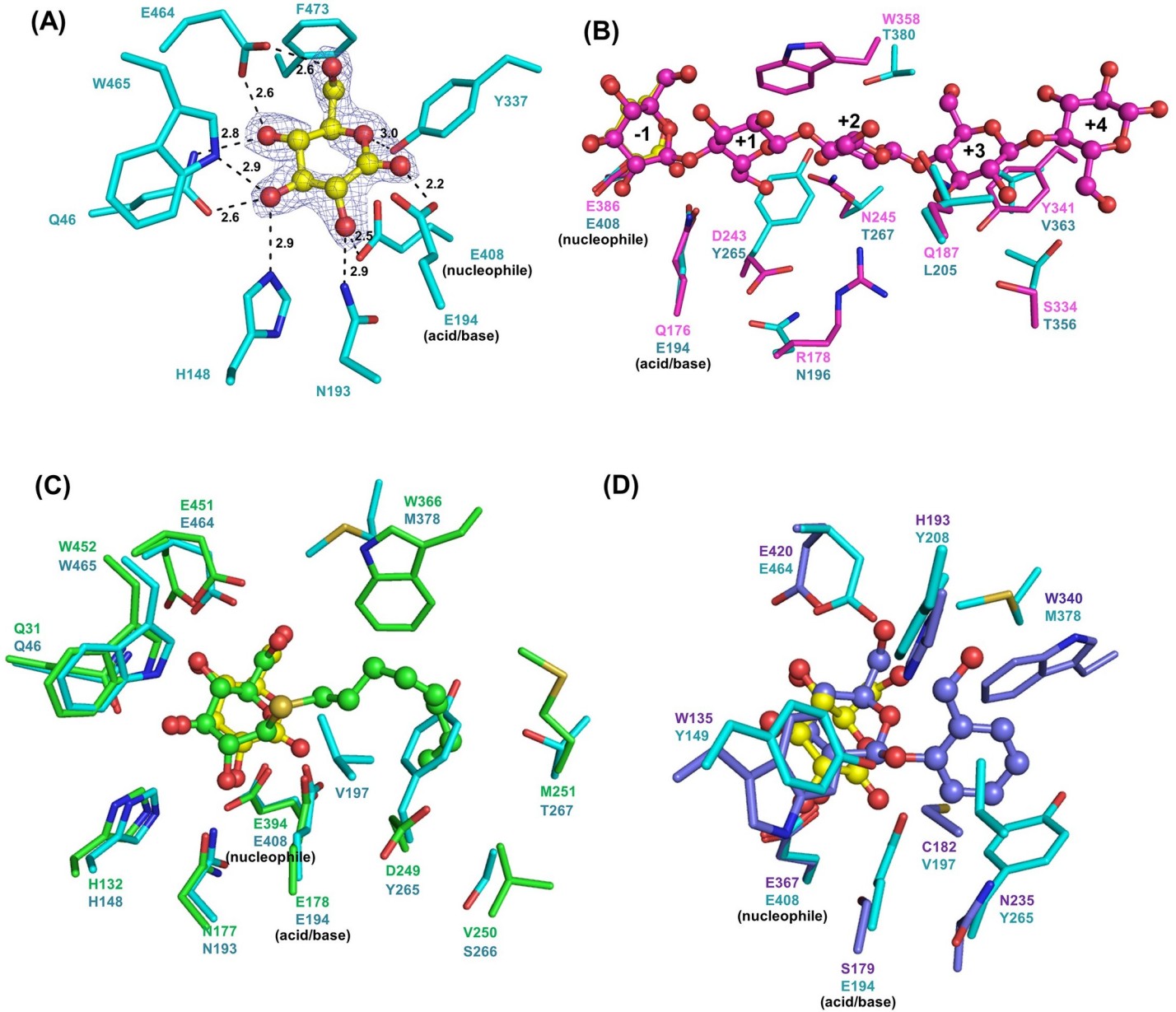

**Fig 2. Comparison of Os4BGlu18 ligand binding with that of other GH1 enzymes.** (A) View of the active site of the Os4BGlu18-DG complex showing protein–ligand interactions. The DG ligand is presented in ball and stick representation, with yellow carbon, while the surrounding amino acid side chains are shown in stick representation with cyan carbons. Oxygen atoms are red and nitrogen atoms blue. The light blue m|$F_o$|–D|$F_c$| omit map for DG is shown contoured at 3 σ. Hydrogen bonds between the protein and the DG are drawn as black dashed lines. (B) The superimposition of Os4BGlu18 with the Os3BGlu7 cellopentaose complex, cellopentaose are indicated in ball and stick. The glucosyl residue-binding subsites observed in the Os3BGlu7 cellopentaose complex are labelled -1, +1, +2, +3 and +4. (C) The superimposition of the Os4BGlu18 with the Os3BGlu6 in complex with *n*-octyl-β-D-thioglucopyranoside. (D) The superimposition of the Os4BGlu18 with the SghA acid/base mutant E179S-salicin complex.

respectively) [52] via hydrogen bonds with the surrounding residues, as shown in Fig 2A. This DG ring was also observed in *Phanerochaete chrysosporium* β-glucosidase [47] and *Neotermes koshunensis* β-glucosidase [48], while an open-form was seen in *Bacillus polymyxa* β-glucosidase [49]. In the active site of Os4BGlu18-DG, the O1 atom of DG forms strong hydrogen bonds to E194 Oε2 (2.3 Å) and E408 Oε1 (3.1 Å). The O2 atom of DG forms hydrogen bonds

to N193 Nε2 (2.9 Å) and E408 Oε2 (2.6 Å). O3 hydrogen bonds with Q46 Oε1 (2.6 Å), H148 Nε2 (2.9 Å), and W465 Nε1 (2.8 Å), while O4 bonded with Q46 Nε2 (2.9 Å) and E464 Oε1 (2.6 Å). DG O5 and O6 form direct hydrogen bonds to the hydroxyl group of Y337 (3.0 Å) and to E464 Oε2 (2.6 Å), respectively. In *P. chrysosporium* β-glucosidase, a significant difference in the E422 side chain position was noted to upon DG binding, which was suggested to be essential for the recognition of the hydroxyl group at O6 of DG [47]. Jeng et al. [48] reported that *N. koshunensis* β-glucosidase in complex with DG (*Nk*Bgl-GNL) structure contained the inhibitor at the glycone-binding site, and a HEPES molecule from the crystallization buffer in the aglycone binding site. However, the superimposition of the native structure of Os4BGlu18 and its DG complex showed that their active sites have very similar amino acid positions and no other molecules were observed at the aglycone binding site.

The superimposition of Os4BGlu18 with rice oligosaccharide exoglucosidase Os3BGlu7 (PDB entry 3f5k) found that there are seven amino acid residues which have polar interactions around cellopentaose are different, including R178, Q187, D243, N245, S334, Y341, and W358 of Os3BGlu7, which are replaced with N196, L205, Y265, T267, T356, V363, and T380 of Os4BGlu18, respectively (Fig 2B) [21]. It has been reported that the active site of Os3BGlu7 is more extensive than the other rice enzymes [21, 39], including Os4BGlu18, and together with the differences in active site cleft residues mentioned above, this may explain why Os4BGlu18 cannot hydrolyze oligosaccharides significantly.

Os4BGlu18 could hydrolyze *n*-octyl-β-D-glucopyranoside, like Os3BGlu6, but with a slower catalytic rate [14]. The structure comparison between Os4BGlu18 and Os3BGlu6 in complex with *n*-octyl-β-D-thioglucopyranoside (PDB entry 3gnp), a *n*-octyl-β-D-glucopyranoside substrate analog [40], is shown in Fig 2C. The amino acid residues that form hydrogen bonds with the glucose in the glycone-binding site are the same, including Q46, H148, N193, E464, and W465 of Os4BGlu18, while the amino acids in the aglycone binding site are different.

Recently, Wang et al. [53] reported the structure of hydrolase SghA, a bacterial virulence factors regulating the infection of plant hosts in *Agrobacterium tumefaciens*, in complex with salicylic acid β-glucoside and salicin. As previously reported [14], Os4BGlu18 also hydrolyzed salicin and superposition of the structures of Os4BGlu18 and the SghA acid/base mutant E179S-salicin complex (PDB entry 6rjo) revealed that most of the residues around salicin were quite similar, except for W135, C182, H193, N235, and W340 of SghA were replaced with Y149, V197, Y208, Y265, and L370 of Os4BGlu18, respectively (Fig 2D). It has been noted that H193 and W340 of SghA might be more important for substrate recognition than other residues. However, the author reported that W340 was conserved in the active site among SghA homologs, while H193 was observed only in the binding cleft of SghA and formed a strong hydrogen bond with the SAG salicylic acid group, but not with salicin.

## Modeling studies on the binding of Os4BGlu18 with monolignol substrates

Several attempts were made to soak monolignols and their glucosides into crystals of Os4B-Glu18 or its E194Q and E194D mutants, but no significant density for these ligands was seen in order to identify residues responsible for specific interactions with monolignol glucosides. Therefore, the natural substrates were docked into the Os4BGlu18 structure, in order to check the interactions of Os4BGlu18 with coniferin, *p*-coumaryl alcohol glucoside, and syringin ligands. Strong binding energies were displayed against these ligands with the ΔG of -8.80, -8.33, and -7.63 kcal/mol, respectively (Table 2). Although *p*-coumaryl alcohol glucoside exhibited higher binding energy than syringin, the biochemical characterization of Os4BGlu18 previously revealed that *p*-coumaryl alcohol glucoside is hydrolyzed at lower catalytic

**Table 2. Molecular docking analysis of Os4BGlu18.**

| Ligand | Binding Energy (kcal mol$^{-1}$) | Amino acid residues forming hydrogen bonds with ligand (within 3.5 Å) | Amino acid with hydrophobic side chains at aglycone binding site within 4 Å |
|---|---|---|---|
| Coniferin | -8.80 | Q46, Y149, N193, E194, Y208, E464, W465, L466 | Y149, Y208, Y265, M378, F381, W465, L466, F467 |
| *p*-coumaryl alcohol glucoside | -8.33 | Q46, Y149, N193, E194, Y208, E464, W465, L466 | Y149, Y208, M378, F381, W465, L466, F467 |
| syringin | -7.63 | Q46, Y149, Y265, Y337, E464, W465 | Y149, V197, Y208, Y265, M378, F381, W465, L466, F467 |

efficiency than syringin [14]. Additionally, the predicted binding energies of long chain oligo-saccharide, cellotetraose and cellopentaose, showed very unfavorable binding energies (3.5 and 87.1 kcal/mol), which corroborates the enzymatic assay results.

The amino acids surrounding the active site cleft that interact with the monolignol gluco-side ligands are noted in Table 2. The residues Q46, Y149, E464 and W465 formed hydrogen bonds with all three ligands, while N193, E194, Y208 and L466 interacted with coniferin and *p*-coumaryl alcohol glucoside and Y265 and Y337 interacted with syringin (Fig 3). At the entrance of the catalytic cleft of Os4BGlu18 are found several amino acid residues with hydro-phobic side chains, including Y149, V197, Y208, Y265, M378, F381, W465, L466 and F473, which further maintain the ligand in the aglycone binding site through pi-pi and hydrophobic interactions. After docking, coniferin makes shorter hydrogen bonds with the catalytic acid/base, E194, and nucleophile, E408, than the other two ligands, which may correlate with the enzymatic study in which Os4BGlu18 prefers coniferin as a substrate [14]. As shown in Fig 3, monolignol substrate ligands adopt a compact conformation of the glucosyl residue at the -1 subsite of the active site of the enzyme, while the monolignols take a part of the aglycone bind-ing pocket. The position and orientation of the glycone part of the three ligands was similar to that of DG in the Os4BGlu18-DG crystal structure. The superposition of the structures con-taining DG and monolignol substrate ligands docked to Os4BGlu18 showed that the interac-tions with the amino acid residues in the glycone-binding pocket (subsite -1) were conserved, thereby validating the docking.

Moreover, molecular dynamics simulation was done to investigate the stability of docked Os4BGlu18 with coniferin over time and the binding energy. The backbone atom position RMSD value was 1.4 Å at the beginning of the production run then gradually reached to about 2.2 Å then settled down to approximately 1.9 Å over the 20 ns simulation, indicating that the studied system was equilibrated over the simulations time (S1 Fig). Investigation of RMSF ver-sus residue showed that the residues at the active site (residue 50 to 240) of the protein were less mobile than the other residues resulting from the stable protein-ligand interaction infer-ring that the ligand was entirely buried in the interior of the active site (S2 Fig). The pair-inter-action energy analysis was done using NAMD Energy plugin between the active site residues (around 3.5 Å from coniferin) and coniferin molecule. The obtaining energy was computed from the non-bonded term including electrostatic energy and van der Waals. The pair-interac-tion energy is plotted over the last 10 ns in S3 Fig. The interaction energy over the last 2 ns of a total 20 ns from MD simulations was -33.36 kcal/mol indicating a high ligand-binding affinity (S3 Fig). Polar interactions with the catalytic amino acids and glycone-binding residues were most frequent over the course of the course of the simulation (S1 Table), suggesting these resi-dues may contribute the most to substrate binding.

Since monolignol β-glucosidases group together in protein sequence-based phylogenetic analysis [14], it was expected that some aglycone-binding residues that may confer monolignol glucoside specificity may be conserved in monolignol β-glucosidases. The conserved residues in monolignol β-glucosidases from rice, Arabidopsis and lodgepole pine observed in multiple

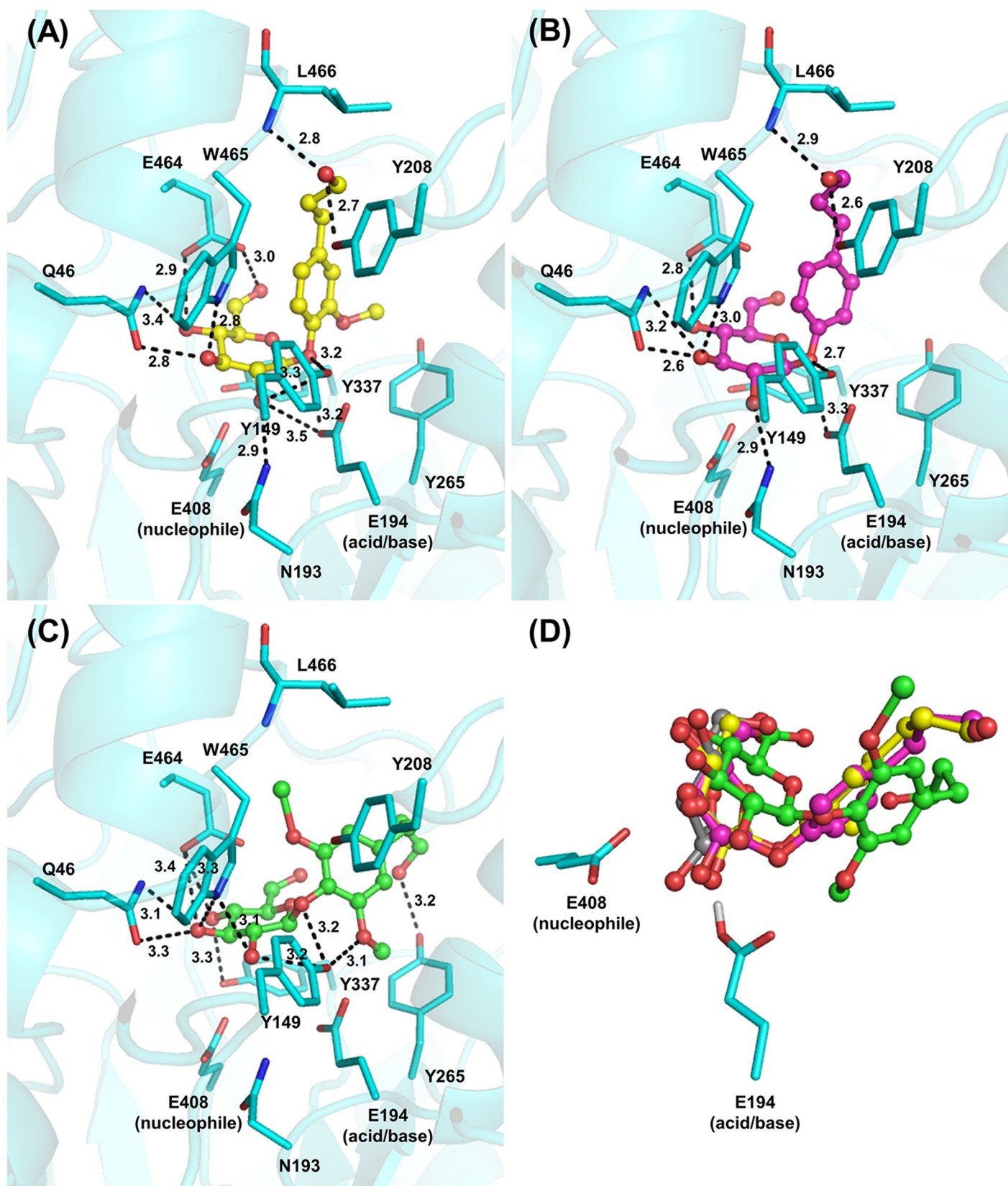

**Fig 3. Molecular docking analysis of rice Os4BGlu18 with its natural substrates ligands.** The interactions between coniferin (A), *p*-coumaryl alcohol glucoside (B), and syringin (C) and the amino acid residues in the active site cleft are shown. Hydrogen bonds (less than 3.5 Å between donor and acceptor atoms) are represented as black dashed lines and the amino acids which make these interactions are shown as sticks. (D) Superimposition of the active sites containing coniferin (yellow carbons), *p*-coumaryl alcohol glucoside (magenta carbons), syringin (green carbons), and glucono δ-lactone (grey carbons) showing that they are in nearly the same position and orientation for the glycone part. Oxygen is shown in red, nitrogen in blue and hydrogen in white.

sequence alignment and homology models of the characterized monolignol β-glucosidases were compared to see what residues may interact with the monolignols in each case (Figs 4 and 5). Rice Os4BGlu14 and Os4BGlu16, Arabidopsis AtBGlu45 and AtBGlu46, and lodgepole pine coniferin β-glucosidase shared 52.45% to 60.30% amino acid sequence identity with Os4BGlu18. The amino acid residues Q46, H148, N193, E194, N335, Y337, E408, W457, E464, W465, and F473 of Os4BGlu18 are conserved in all β-glucosidases and are involved in glycone binding. In contrast, the residues Y149, V197, H201, Y208, Y265, M378, T380, and F381 that are positioned around the aglycone binding region were remarkably variable, even within monolignol beta-glucosidases. The aromatic amino acids Y149 and Y208 were conserved for Os4BGlu18 and Os4BGlu14, while the other monolignol β-glucosidases have the conserved change to F at that those positions. Interestingly, Y149 is replaced by V in AtBGLU45 and W in most other plant β-glucosidases with x-ray crystal structures. The amino acid in the Y208 position varies among other GH1 β-glucosidases, some of which also have F or Y in this position. So, although Y149 and Y208 may contribute to specificity for monolignols in Os4BGlu18, only the aromatic residue at the 208 position is conserved among other known monolignol β-glucosidase. Moreover, M378, T380 and F381 located at the aglycone binding site differed among monolignol β-glucosidases, although the M378 position was hydrophobic in monolignol β-glucosidases and polar in the other plant β-glucosidases compared. Residues V197, H201, and Y265 were also dissimilar. Based on the results of molecular docking and homology modeling, each of the monolignol β-glucosidases could provide aromatic interactions with the phenolic ring of the monolignols, as well as hydrophobic groups to interact with the nonpolar tail and polar groups to interact with the hydroxyl, but few of these interactions were conserved.

## Conclusions

In conclusion, we successfully crystallized and solved the structures of Os4BGlu18 and Os4B-Glu18 in complex with DG at 1.7 and 2.1 Å resolution, respectively. The overall structure of Os4BGlu18 was nearly the same as that of rice Os3BGlu7, used for the molecular replacement template, and other known GH1 structures, but significant differences were found in four variable loops. Os4BGlu18 contains two disulfide bridges as observed in the Os4BGlu12, which could affect enzyme stabilization [41]. The DG in the Os4BGlu18-DG complex was found in the ring-form and made hydrogen bonds with Q46, H148, N193, E194, Y337, E408, E464, and W465, which are conserved glucose-binding residues found in GH1 β-glucosidases. The superimposition of Os4BGlu18 and Os3BGlu7 revealed different amino acid residues in the aglycone-binding pocket and the active site shape, consistent with their different substrate specificities.

Coniferin, *p*-coumaryl alcohol glucoside, and syringin docked into the Os4BGlu18 displayed strong binding energies, while binding of cellotetraose and cellopentaose was unfavorable, consistent with the enzymatic activity of Os4BGlu18. The aglycone interacted primarily with aromatic and nonpolar groups from residues, such as Y149, Y208, M378, and F381, reflecting the nonpolar nature of the substrates, with a few hydroxyls, such as on T380 to accommodate the terminal hydroxyl. Interestingly, while other monolignol β-glucosidases have similar overall sequences and conserved sugar-binding residues, along with similar

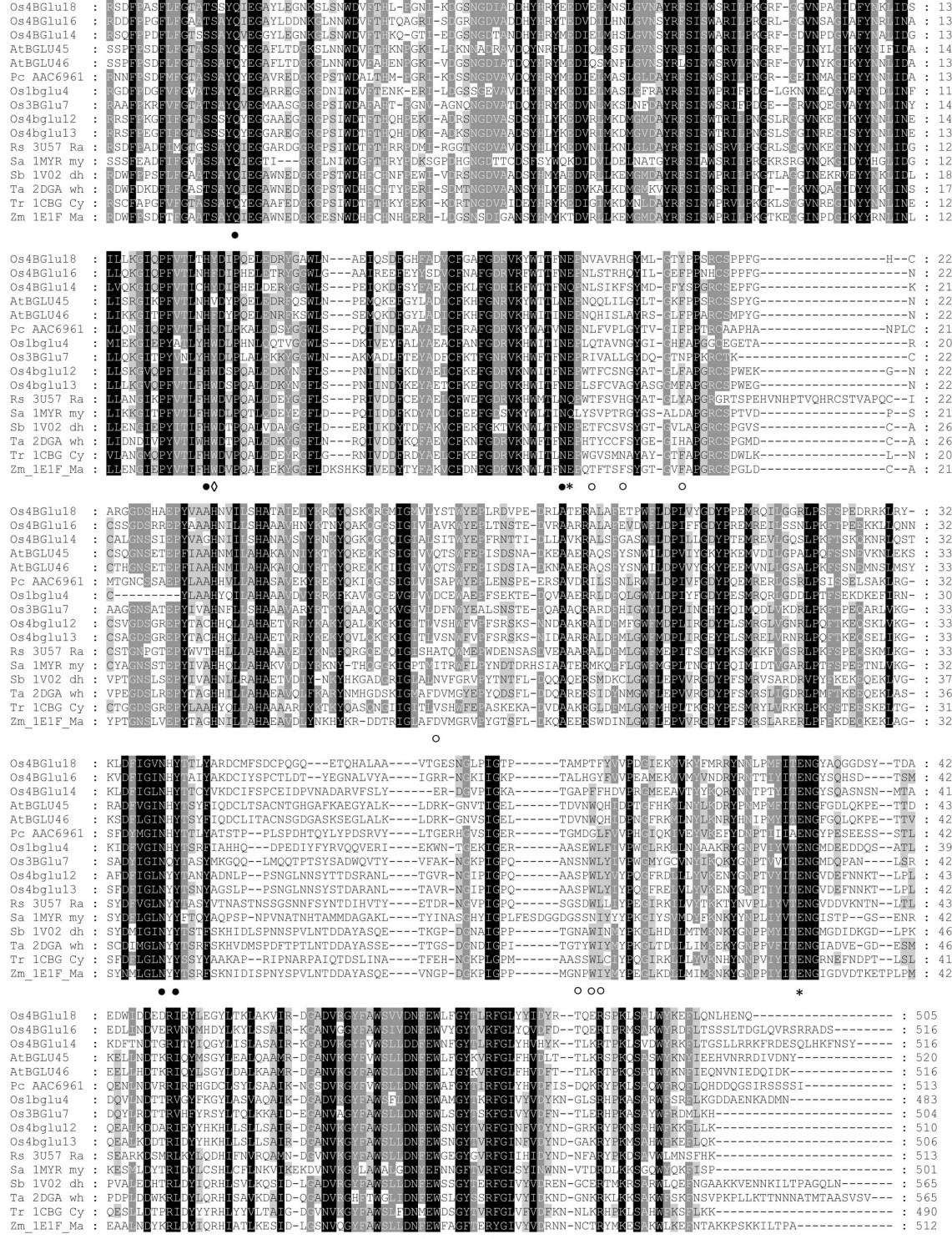

**Fig 4. Multiple sequence alignment of characterized monolignol β-glucosidases and other plant β-glucosidases.** The monolignol β-glucosidase are containing *Oryza sativa* (Os4BGlu14, Os4BGlu16, and Os4BGlu18), *Arabidopsis thaliana* (AtBGLU45 and AtBGLU46), and *Pinus contorta* (PC_AAC6961). Rice β-glucosidases consist of Os1BGlu4, Os3BGlu7, Os4BGlu12, and Os4BGlu13. The raucaffricine-*O*-β-D-glucosidase of *Rauvolfia serpentine* is labeled as Rs_3U57_Ra. Sa_1MYR_my is *Sinapis alba* myrosinase; Sb_1V02_dh is *Sorghum bicolor* dhurrinase; Ta_2DGA_wh is *Triticum aestivum* β-glucosidase; Tr_1CBG_Cy is *Trifolium repens* cyanogenic β-glucosidase; and Zm_1E1F_Ma is *Zea mays* β-glucosidase. The alignment was generated by the MUSCLE algorithm implemented in MEGA-X. Stars represent the catalytic acid/base and nucleophilic residues, filled circles indicate conserved amino acid

residues around the glycone-binding site of all β-glucosidase, opened circles show the corresponding residues at aglycone-binding of Os4BGlu18 and diamonds denote aromatic residues that may interact with the monolignol aromatic ring found in (nearly) all monolignol β-glucosidases.

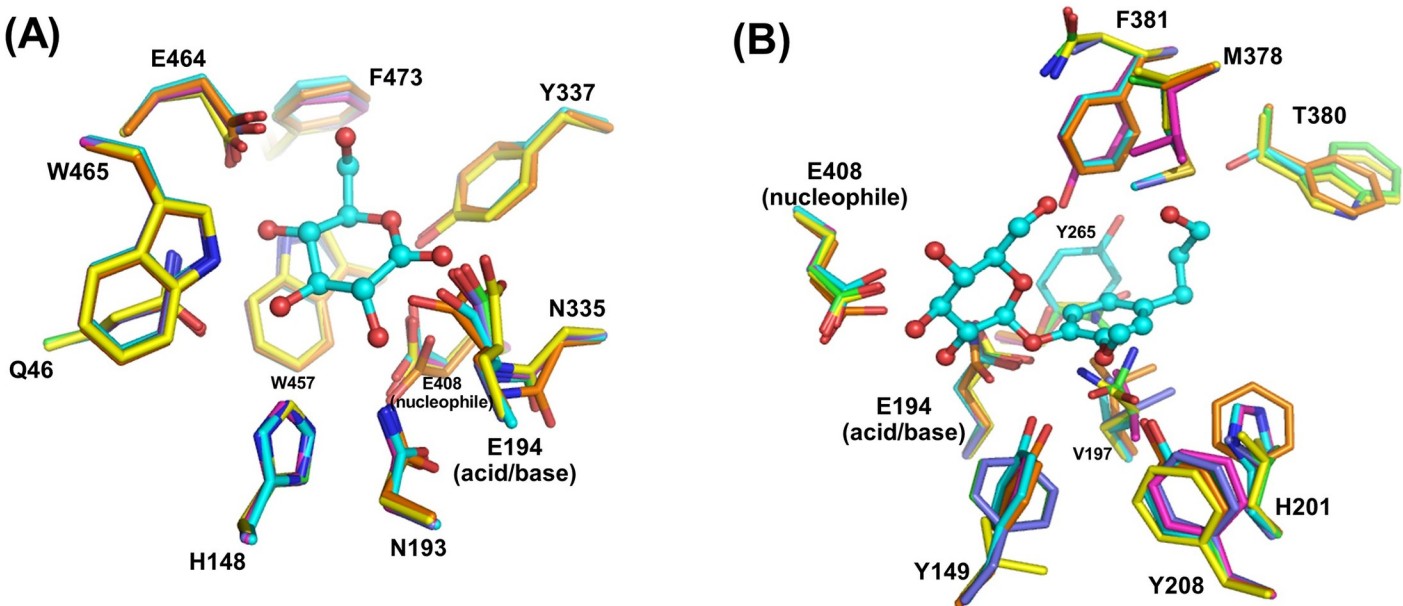

**Fig 5. Glycone and aglycone binding residues of the monolignol β-glucosidases.** (A-B) Superimposition of the Os4BGlu18-DG complex and docked Os4BGlu18-coniferin with other characterized monolignol β-glucosidase structures generated by homology modeling. Rice Os4BGlu14 is shown with orange, Os4BGlu16 with magenta, Os4BGlu18 with cyan, Arabidopsis AtBGlu45 with yellow, AtBGlu46 with green, and lodgepole pine coniferin β-glucosidase with slate carbon atoms. Oxgen is shown in red, nitrogen in blue and sulfur in yellow.

functional groups to bind the substrates, few of the aglycone residues are conserved. Thus, although monolignol β-glucosidases are grouped together by sequence-based phylogenetic analysis [13–15] and fulfill the same function, they achieve aglycone-specificity by unexpectedly diverse means.

## Supporting information

**S1 Fig. RMSD of protein backbone compared with the initial structure over 20-ns.** (TIF)

**S2 Fig. RMSF per residue upon ligand binding.** (TIF)

**S3 Fig. Time course of pair-interaction energy between protein and coniferin.** (TIF)

**S4 Fig. A structure of coniferin with oxygen atoms number which make hydrogen bonds with the protein numbered according to their numbers in S1 Table.** (TIF)

**S1 Table. Hydrogen bond analysis.** Hydrogen bond calculation was analyzed with a distance cutoff of 3.5 Å and an angle cutoff of 30˚. Forty-six hydrogen bonds were found as listed. The numbering of the oxygen atoms is indicated in S4 Fig. (DOCX)

**S1 Report.**
(PDF)

**S2 Report.**
(PDF)

## Acknowledgments

Genji Kurisu and Ratana Charoenwattanasatien are thanked for facilitating data collection at the SPring-8 synchrotron, while the SPring-8 beamline BL44XU and National Synchrotron Radiation Research Center (NSRRC) beamline 15A staff are thanked for their support and help during data collection.

## Author Contributions

**Conceptualization:** Supaporn Baiya, James R. Ketudat Cairns.

**Data curation:** Supaporn Baiya, Salila Pengthaisong.

**Formal analysis:** Supaporn Baiya, Salila Pengthaisong, James R. Ketudat Cairns.

**Funding acquisition:** James R. Ketudat Cairns.

**Investigation:** Supaporn Baiya, Sunan Kitjaruwankul.

**Methodology:** Salila Pengthaisong, Sunan Kitjaruwankul.

**Project administration:** James R. Ketudat Cairns.

**Resources:** James R. Ketudat Cairns.

**Supervision:** James R. Ketudat Cairns.

**Validation:** Supaporn Baiya, Salila Pengthaisong, Sunan Kitjaruwankul.

**Visualization:** Supaporn Baiya.

**Writing – original draft:** Supaporn Baiya.

**Writing – review & editing:** Salila Pengthaisong, Sunan Kitjaruwankul, James R. Ketudat Cairns.

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
