## [Decision Letter · Decision Letter 0]

2 Nov 2020

PONE-D-20-31981

Structural analysis of rice Os4BGlu18 monolignol β-glucosidase

PLOS ONE

Dear Dr. Cairns,

Thank you for submitting your manuscript to PLOS ONE. After careful consideration, we feel that it has merit but does not fully meet PLOS ONE’s publication criteria as it currently stands. Therefore, we invite you to submit a revised version of the manuscript that addresses the points raised during the review process.

In the revised version you are requested to address as fully as possible the constructive comments of both reviewers.

We look forward to receiving your revised manuscript.

Kind regards,

Israel Silman

Academic Editor

PLOS ONE

Journal Requirements:

Reviewers' comments:

Reviewer's Responses to Questions

**Comments to the Author**

1. Is the manuscript technically sound, and do the data support the conclusions?

Reviewer #1: Partly

Reviewer #2: Yes

2. Has the statistical analysis been performed appropriately and rigorously? 

Reviewer #1: N/A

Reviewer #2: Yes

3. Have the authors made all data underlying the findings in their manuscript fully available?

Reviewer #1: Yes

Reviewer #2: Yes

4. Is the manuscript presented in an intelligible fashion and written in standard English?

Reviewer #1: Yes

Reviewer #2: Yes

5. Review Comments to the Author

Reviewer #1: The manuscript submitted by Baiyna and coworkers shows the determination of the crystal structure of the rice beta-glusidase Os4BGlu18, together with the complex of the same enzyme with glucolactone. The structures were determined at good resolutions, allowing the detailed mapping of the differences between this enzyme and other GH1 beta-glucosidases known to date. The manuscript also describes the molecular docking of monolignol substrates, previously known to be hydrolyzed by Os4BGlu18, and on the basis of the docked poses, the authors describe the atomic details of the recognition between the monolignols and the enzyme.

The manuscript is concise and very well written. I also acknowledge that the authors made a very desirable effort to cite many important papers in the field of beta-glucosidases structural biology.

As my major concern, I would say that the major finding I see in the manuscript is the description of the crystal structures. Maybe, the manuscript would be more appropriate for a journal in the field, such as Acta CrystD or Acta CrystF. However, I leave this decision to the editor.

That said, I will list some (minor) issues that could be revisited by the authors to improve the clarity:

1. Crystal structures. The Table 1 shows very low RMS deviations for bonds, even for crystal structures that were determined for a relatively high resolution. Can the authors comment something on that?

2. Also, for the details of the crystal structure shown in Table1, I found it curious that the Os4BGlu18-DG complex has 0.5% of its residues in the disallowed region of the Ramachandran diagram. Molprobity would flag it as a warning, for example. It would be nice if the authors could comment/revisit their refinement details.

3. The details provided in the docking methodology could be extended to include some additional details. I am not familiar with Discovery Studio, but I understood that the software somehow deals with sugars appropriately. In the other hand, the details for the preparation of the enzyme are not given in the manuscript. I *assume* that the authors used a force field-based energy optimization in the Lamarckian optimization in AutoDock. For this purpose, the authors had to decide on the protonation states for the residues in the protein. The most important residue to look at is the acid/base glutamic acid, that should be protonated in the active site. This matter, since many programs would automatically assign the residue to be deprotonated and charged. How did the authors deal with the receptor preparation? Figure 3D suggests that the residue was deprotonated.

4. Even global optimization algorithms can sometimes be stuck in local minima in the context of the docking problem. The problem can get even more tricky when the enzyme structure was not solved with a ligand in its active site (so that the structure is not ‘adapted’ to ligand binding). In this context, I have to issues:

a. Why did the authors decide to use the native structure instead of the crystal structure bound to glucolactone? Is there any particular reason for choosing this crystal structure?

b. Is it possible for the authors to run a MD simulation, for example, to test whether the docked conformations are stable over time? It would also be nice to analyze the interaction energies for the MD ensemble instead of the single conformation from docking.

5. (A comment, more than a question): It is interesting to see that a MES molecule bound close to the active site but not exactly in the active site. The finding published by Jeng and coworkers came right at the moment to my mind, and the authors discussed those findings as well.

6. It is also interesting that a Zinc atom is placed between the two monomers in the ASU. Is there any experimental evidence of Zinc-mediated dimerization for Os4BGlu18 ?

7. Figure 1C: It would be nice if the interacting distances could be shown in the picture. Same for Figure 2A and Figure 3.

Reviewer #2: This manuscript describes three-dimensional structure determination and docking analysis of a GH1 beta-glucosidase from rice (Os4Bglu18). The authors have been previously demonstrated that this enzyme is specifically active for monolignol beta-glucosides and can functionally rescue the A. thaliana bglu45 mutant (reference 12). The crystal structure was determined in both the ligand-free and the delta-gluconolactone (DG) complex forms with good quality. The docking study was reasonably done, and it revealed the protein-substrate interaction of the aglycone part. The manuscript is concisely and clearly written. Because this is the first report of the three-dimensional structure of monolignol beta-glucosidases, it will attract readers’ attention in the context of lignin biosynthesis of the plant cell wall. I have some suggestions to further improve this work.

Major points:

1. Table 1. If the CC1/2 values correspond to the highest resolution (outer) shell, they should be in parentheses. The resolution cutoff of these data (especially for the DG complex) appears to be too “stoic” for me. If the authors can re-process the raw diffraction image data that include diffractions at higher angles, they may reach higher resolutions. I think CC1/2 > 0.6 at the outer shell is reasonably allowed recently.

2. L115-124. The methodology of automated docking should be described in detail. I think AutoDockTools is just a GUI software. Authors should describe what version of the docking software (AutoDock 4.x or AutoDock Vina?) was used.

3. L147. If the crystallization solution did not include zinc ion, was the metal co-purified from the E. coli extract? The CheckMyMetal server will support the metal assignment.

4. L160-162. The disulfide bonds should be shown in Fig. 1.

5. L188-208. Describe the sugar ring conformation of DG with the Cremer-Pople parameter.

6. Table 2. Units (kcal/mol) of the binding energy must be written. I think the positive energy values (3.50 and 87.11) do not make sense and are not appropriate for putting on the table.

7. Fig. 2. For comparison with other beta-glucosidases, the DG molecule in Os4Bglu18 should be also displayed.

8. Side chains of some key residues for the aglycone recognition (e.g., Y208 and M378?) can be moved during the autodocking search. That may give better docking estimation or may result in divergence due to an increased degree of freedom. But it is worth trying.

Minor points:

1. L44-47. Please check the grammar of this sentence.

2. The protein sample for crystallization was diluted to 3 mg/ml (L82), and the crystallization setup was done with protein concentrations of 1-8 mg/ml (L87).

3. L95. Spring-8 -> SPring-8

4. L103. Refinement statistic(s) are ..

5. L241. ”‖Fo|–|Fc‖ omit map”. I think modern crystallographic tools usually make mFo-DFc omit maps or something like that in default.

6. PLOS authors have the option to publish the peer review history of their article (what does this mean?). If published, this will include your full peer review and any attached files.

Reviewer #1: **Yes: **Alessandro S. Nascimento

Reviewer #2: No

---

## [Author Response · Author response to Decision Letter 0]

16 Dec 2020

Reviewer #1: The manuscript submitted by Baiya and coworkers shows the determination of the crystal structure of the rice beta-glusidase Os4BGlu18, together with the complex of the same enzyme with glucolactone. The structures were determined at good resolutions, allowing the detailed mapping of the differences between this enzyme and other GH1 beta-glucosidases known to date. The manuscript also describes the molecular docking of monolignol substrates, previously known to be hydrolyzed by Os4BGlu18, and on the basis of the docked poses, the authors describe the atomic details of the recognition between the monolignols and the enzyme.

The manuscript is concise and very well written. I also acknowledge that the authors made a very desirable effort to cite many important papers in the field of beta-glucosidases structural biology.

As my major concern, I would say that the major finding I see in the manuscript is the description of the crystal structures. Maybe, the manuscript would be more appropriate for a journal in the field, such as Acta CrystD or Acta CrystF. However, I leave this decision to the editor.

That said, I will list some (minor) issues that could be revisited by the authors to improve the clarity:

1. Crystal structures. The Table 1 shows very low RMS deviations for bonds, even for crystal structures that were determined for a relatively high resolution. Can the authors comment something on that?

Response: When refining the structures we use a standard of RMSD bond of 0.014 Å or less and RMSD bond-angles of 1.4 degree or less to set the weighting. In these cases, the restraints to keep the bond angle below 1.4 degrees resulted in lower bond length RMSD values (0.007-0.008 Å). Relaxing it further results in only minor improvement in the Rfree values.

2. Also, for the details of the crystal structure shown in Table1, I found it curious that the Os4BGlu18-DG complex has 0.5% of its residues in the disallowed region of the Ramachandran diagram. Molprobity would flag it as a warning, for example. It would be nice if the authors could comment/revisit their refinement details.

Response: The values provided in Table 1 were generated with PROCHECK. Molprobity gives much better Ramachandran statistics. All 4 amino acid residues in the 0.5% in the PROCHECK report fit to the electron density, but they are in the disallowed region of the Ramachandran diagram in PROCHECK. Only one, Asp416 of Mol B, was in the disallowed regions of the Molprobity and PDB validation reports (as noted above, it fit the density well). If desired, we could replace the PROCHECK values with the Molprobity values from the PDB validation report, but we kept the more stringent values for now.

3. The details provided in the docking methodology could be extended to include some additional details. I am not familiar with Discovery Studio, but I understood that the software somehow deals with sugars appropriately. In the other hand, the details for the preparation of the enzyme are not given in the manuscript. I *assume* that the authors used a force field-based energy optimization in the Lamarckian optimization in AutoDock. For this purpose, the authors had to decide on the protonation states for the residues in the protein. The most important residue to look at is the acid/base glutamic acid, that should be protonated in the active site. This matter, since many programs would automatically assign the residue to be deprotonated and charged. How did the authors deal with the receptor preparation? Figure 3D suggests that the residue was deprotonated.

Response: We have increased the description of the docking a little and have rerun it with the catalytic acid/base protonated, as suggested by the reviewer. The polar hydrogen and Kollman charges, then the specific hydrogen was manually added to the catalytic acid/base in the pdbqt file and we then ran AutoDock again.

4. Even global optimization algorithms can sometimes be stuck in local minima in the context of the docking problem. The problem can get even more tricky when the enzyme structure was not solved with a ligand in its active site (so that the structure is not ‘adapted’ to ligand binding). In this context, I have to issues:

a. Why did the authors decide to use the native structure instead of the crystal structure bound to glucolactone? Is there any particular reason for choosing this crystal structure?

Response: The superposition of free structure and complex with gluconolactone was quite similar, even in the active site.

b. Is it possible for the authors to run a MD simulation, for example, to test whether the docked conformations are stable over time? It would also be nice to analyze the interaction energies for the MD ensemble instead of the single conformation from docking.

Response: We have run a 20-ns MD simulation and found that the ligand was entirely buried in the interior of the active site throughout the simulation. We have added this description to the end of the Results (in addition to adding the methods in the Methods section). The data from the MD are shown in the S1-S4 figures and S1 Table.

5. (A comment, more than a question): It is interesting to see that a MES molecule bound close to the active site but not exactly in the active site. The finding published by Jeng and coworkers came right at the moment to my mind, and the authors discussed those findings as well.

Response: Thank you for the comment. We often find MES in beta-glucosidase active sites, for some reason. Perhaps it has a good mixture of polar and nonpolar properties for binding those sites.

6. It is also interesting that a Zinc atom is placed between the two monomers in the ASU. Is there any experimental evidence of Zinc-mediated dimerization for Os4BGlu18 ?

Response: It was confirmed by CheckMyMetal server with no alternative metals as the attach file CMM_output.pdf. However, our group previously published Os3BGlu7 and Os4BGlu12 structures and they were also included zinc with similar preparation without intentionally adding zinc, suggesting that it could be contaminated in the salt. In Os3BGlu7 (BGlu1), the assignment as Zn was supported by X-ray fluorescence (Chuenchor et al., 2008). We have added the verification with CheckMyMetal to the paper, along with the possible explanation of contamination of the NaCl (generally, Zn is listed in the minor contaminants). 

7. Figure 1C: It would be nice if the interacting distances could be shown in the picture. Same for Figure 2A and Figure 3.

Response: The distances between polar interacting groups have been added to Figures 1C, 2A and 3 A, B and C, as suggested.

Reviewer #2: This manuscript describes three-dimensional structure determination and docking analysis of a GH1 beta-glucosidase from rice (Os4Bglu18). The authors have been previously demonstrated that this enzyme is specifically active for monolignol beta-glucosides and can functionally rescue the A. thaliana bglu45 mutant (reference 12). The crystal structure was determined in both the ligand-free and the delta-gluconolactone (DG) complex forms with good quality. The docking study was reasonably done, and it revealed the protein-substrate interaction of the aglycone part. The manuscript is concisely and clearly written. Because this is the first report of the three-dimensional structure of monolignol beta-glucosidases, it will attract readers’ attention in the context of lignin biosynthesis of the plant cell wall. I have some suggestions to further improve this work.

Major points:

1. Table 1. If the CC1/2 values correspond to the highest resolution (outer) shell, they should be in parentheses. The resolution cutoff of these data (especially for the DG complex) appears to be too “stoic” for me. If the authors can re-process the raw diffraction image data that include diffractions at higher angles, they may reach higher resolutions. I think CC1/2 > 0.6 at the outer shell is reasonably allowed recently.

Response: We have corrected the CC1/2 as suggested. We cannot use higher resolution data due to the incompleteness at higher resolutions. 

2. L115-124. The methodology of automated docking should be described in detail. I think AutoDockTools is just a GUI software. Authors should describe what version of the docking software (AutoDock 4.x or AutoDock Vina?) was used.

Response: We have corrected it as suggested: “Molecular docking was performed using Autodock 4.2 (ADT version 1.5.6) to investigate the interactions of Os4BGlu18 with its natural substrates. All hetero atoms and molecule B residues were deleted from the native Os4BGlu18 structure. Polar hydrogen atoms and Kollman charges were added to the protein. A hydrogen atom was manually added to the catalytic acid/base , E194, to be consistent with its role in protonating the glycosidic oxygen in the first step of catalysis. The p-coumaryl alcohol glucoside, coniferin, and syringin ligands were prepared with the glucosyl moiety in the 1S3 skew boat conformation with Discovery Studio 4.0 program (Dassault Systèmes BIOVIA, San Diego, CA) and set the number of active torsions as 6, 7, and 8, respectively. The aromaticity criterion was set to be 7.5. The ligands were docked into the active site of Os4BGlu18 with the grid box dimension 60x60x60 points in x, y, and z and a grid spaceing of 0.375 Å via a Lamarkian genetic algorithm methodology. The docking was run 200 times for each substrate using Cygwin and the conformation which showed the best binding energy (ΔG) was selected [40].”

3. L147. If the crystallization solution did not include zinc ion, was the metal co-purified from the E. coli extract? The CheckMyMetal server will support the metal assignment.

Response: It was confirmed by CheckMyMetal server with no alternative metals as the attach file CMM_output.pdf. In addition, our group previously published Os3BGlu7 and Os4BGlu12 structures and they were also included zinc with similar preparation, suggesting that it could be contaminated in the salt. We have added the CheckMyMetal verification and the possible explanation of contamination of the NaCl to the paper.

4. L160-162. The disulfide bonds should be shown in Fig. 1.

Response: The disulfide bonds are now shown in Fig. 1B.

5. L188-208. Describe the sugar ring conformation of DG with the Cremer-Pople parameter.

Response: We have added the conformation of the DG and its Cremer-Pople parameters to the description: “The Os4BGlu18-DG complex structure contains DG in the ring-form in a 4H3/4E conformation (Cremer-Pople parameters: φ (°), θ (°), Q: 224.521, 47.426, 0.534 and 227.030, 44.337, 0.524 in molecules A and B, respectively) [52] via hydrogen bonds with the surrounding residues…” 

6. Table 2. Units (kcal/mol) of the binding energy must be written. I think the positive energy values (3.50 and 87.11) do not make sense and are not appropriate for putting on the table.

Response: We have corrected it as suggested, including deleting the docking with unfavorable values from the table.

7. Fig. 2. For comparison with other beta-glucosidases, the DG molecule in Os4Bglu18 should be also displayed.

Response: We have corrected it as suggested.

8. Side chains of some key residues for the aglycone recognition (e.g., Y208 and M378?) can be moved during the autodocking search. That may give better docking estimation or may result in divergence due to an increased degree of freedom. But it is worth trying.

Response: We have tried to run AutoDock with the flexible amino acid sidechains at the aglycone site of protein as suggested, but the result showed that they moved to unrealistic positions, since in our experience, there are few changes in the positions of GH1 enzymes in complexes with substrates and inhibitors.

Minor points:

1. L44-47. Please check the grammar of this sentence.

Response: We have corrected the sentence as follows: “It is a natural polyphenolic polymer composed of the p-hydroxyphenyl (H), guaiacyl (G), and syringyl (S) units, which are derived from oxidative polymerization of the corresponding monolignols, p-coumaryl alcohol, coniferyl alcohol, and sinapyl alcohol, respectively [2].”

2. The protein sample for crystallization was diluted to 3 mg/ml (L82), and the crystallization setup was done with protein concentrations of 1-8 mg/ml (L87).

Response: The 3 mg/ml protein was used for microbatch screening test to find the best precipitants from the crystal screen HT kit, while 1-8 mg/ml was used for optimization of protein concentration in hanging drop vapor diffusion method. We tried to clarify this point by adjusting the first sentence as follows: “To screen for Os4BGlu18 protein crystallization conditions, the protein was diluted to 3 mg/ml with a buffer containing 20 mM Tris-HCl, pH 8.0, 150 mM NaCl.”

3. L95. Spring-8 -> SPring-8

Response: We have corrected it as suggested. 

4. L103. Refinement statistic(s) are ..

Response: We have corrected it as suggested.

5. L241. ”‖Fo|–|Fc‖ omit map”. I think modern crystallographic tools usually make mFo-DFc omit maps or something like that in default.

Response: Yes, indeed, the maps are the weighted forms (mFo-DFc), but we just used the Fo-Fc designation as a shorthand, as found in many papers. To be clear, we have changed it as indicated.

---

## [Decision Letter · Decision Letter 1]

29 Dec 2020

Structural analysis of rice Os4BGlu18 monolignol β-glucosidase

PONE-D-20-31981R1

Dear Dr. Cairns,

We’re pleased to inform you that your manuscript has been judged scientifically suitable for publication and will be formally accepted for publication once it meets all outstanding technical requirements.

Kind regards,

Israel Silman

Academic Editor

PLOS ONE

Additional Editor Comments (optional):

Reviewers' comments:

Reviewer's Responses to Questions

**Comments to the Author**

1. If the authors have adequately addressed your comments raised in a previous round of review and you feel that this manuscript is now acceptable for publication, you may indicate that here to bypass the “Comments to the Author” section, enter your conflict of interest statement in the “Confidential to Editor” section, and submit your "Accept" recommendation.

Reviewer #1: All comments have been addressed

Reviewer #2: All comments have been addressed

2. Is the manuscript technically sound, and do the data support the conclusions?

Reviewer #1: Yes

Reviewer #2: Yes

3. Has the statistical analysis been performed appropriately and rigorously? 

Reviewer #1: N/A

Reviewer #2: Yes

4. Have the authors made all data underlying the findings in their manuscript fully available?

Reviewer #1: Yes

Reviewer #2: Yes

5. Is the manuscript presented in an intelligible fashion and written in standard English?

Reviewer #1: Yes

Reviewer #2: Yes

6. Review Comments to the Author

Reviewer #1: The authors have properly addressed the issued pointed by the reviewers. Some issues related to the structure refinement were answered, the methods used for the ligand docking were better described, allowing other researchers to reproduce the authors’ results and the (short) MD simulation introduces some equilibrium analysis to the substrate-enzyme interaction analysis.

In this reviewer’s view, the manuscript is suitable for publication in its current form.

Alessandro S. Nascimento

Reviewer #2: I just want to add one comment.

Please mind the significant figures of the Cremer-Pople parameters.

I think ~3 digits should be better.

7. PLOS authors have the option to publish the peer review history of their article (what does this mean?). If published, this will include your full peer review and any attached files.

Reviewer #1: **Yes: **Alessandro S. Nascimento

Reviewer #2: No

---

## [Editor Report · Acceptance letter]

11 Jan 2021

PONE-D-20-31981R1 

Structural analysis of rice Os4BGlu18 monolignol β-glucosidase 

Dear Dr. Ketudat Cairns:

I'm pleased to inform you that your manuscript has been deemed suitable for publication in PLOS ONE. Congratulations! Your manuscript is now with our production department. 

Kind regards, 

on behalf of

Prof. Israel Silman 

Academic Editor

PLOS ONE